# EMG-Controlled Soft Robotic Bicep Enhancement [note 1]

**DOI:** 10.3390/bioengineering12050526

**Published:** 2025-05-15

**Authors:** Jiayue Zhang, Daniel Vanderbilt, Ethan Fitz, Janet Dong

**Affiliations:** Department of Mechanical and Materials Engineering, University of Cincinnati, Cincinnati, OH 45221-0072, USA; zhang3je@mail.uc.edu (J.Z.); vanderdi@mail.uc.edu (D.V.); fitzed@mail.uc.edu (E.F.)

**Keywords:** McKibben pneumatic muscle, soft robotics, EMG sensor, PID control

## Abstract

Industrial workers often engage in repetitive lifting tasks. This type of continual loading on their arms throughout the workday can lead to muscle or tendon injuries. A non-intrusive system designed to assist a worker’s arms would help alleviate strain on their muscles, thereby preventing injury and minimizing productivity losses. The goal of this project is to develop a wearable soft robotic arm enhancement device that supports a worker’s muscles by sharing the load during lifting tasks, thereby increasing their lifting capacity, reducing fatigue, and improving their endurance to help prevent injury. The device should be easy to use and wear, functioning in relative harmony with the user’s own muscles. It should not restrict the user’s range of motion or flexibility. The human arm consists of numerous muscles that work together to enable its movement. However, as a proof of concept, this project focuses on developing a prototype to enhance the biceps brachii muscle, the primary muscle involved in pulling movements during lifting. Key components of the prototype include a soft robotic muscle or actuator analogous to the biceps, a control system for the pneumatic muscle actuator, and a method for securing the soft muscle to the user’s arm. The McKibben-inspired pneumatic muscle was chosen as the soft actuator for the prototype. A hybrid control algorithm, incorporating PID and model-based control methods, was developed. Electromyography (EMG) and pressure sensors were utilized as inputs for the control algorithms. This paper discusses the design strategies for the device and the preliminary results of the feasibility testing. Based on the results, a wearable EMG-controlled soft robotic arm augmentation could effectively enhance the endurance of industrial workers engaged in repetitive lifting tasks.

## 1. Introduction

Repeated heavy lifting of objects by industrial workers throughout the workday can lead to serious arm injuries, such as muscle strains and tendon injuries. A report from the U.S. Bureau of Labor Statistics (BLS) [1], in 2022, revealed that there were 159,700 cases of musculoskeletal disorders affecting the upper extremities, with an incidence rate of eight cases per 10,000 full-time workers. The median time away from work due to these injuries is 24 days, resulting in significant downtime for workers and negatively impacting their productivity.

This research aims to address these issues by developing a personal protective device. The device will be wearable, compact, lightweight, and easy to use in manufacturing settings. Its purpose is to reduce the likelihood of arm injuries by increasing arm endurance and enhancing arm strength.

Currently, exoskeletons are widely used in various industries because they provide excellent support for the upper body and arms. However, they tend to be bulky and heavy and difficult for individual workers to wear and fit properly. Our research focuses on a different strategy: using soft muscles instead of rigid structural components to address arm injury issues.

So far, scientists and engineers have developed several soft actuation mechanisms for various applications, aiming to enhance human strength. They include Dielectric Elastomer Actuators [2], Relaxor Ferroelectric Polymer Actuators (magnetism) [3], Liquid Crystal Elastomers [4], Conducting Polymer Actuators [5,6], Polymeric Molecular Actuators [7], Ionic Polymer/Metal Composites [8], Carbon Nanotube Actuators [9], Thermally Activated Shape Memory Alloys [10], Ferromagnetic Shape Memory Alloys [11], McKibben-inspired pneumatic muscles [12], and fishing line muscles [13]. Among these soft actuators, the two that are most widely used and producible are the pneumatic muscle and the fishing line muscle. Both artificial muscles achieve linear motion through the elongation and contraction of their soft materials. The advantages of the fishing line muscle include its lightweight design, compact size, and high force-to-weight ratio. However, the function of fishing line muscles relies on manipulating nylon’s properties through precise temperature control [14], necessitating complex control methods that lack the robustness and responsiveness needed for industrial applications. The McKibben-inspired pneumatic muscle is straightforward and cost effective to construct. It responds quickly and provides high stability. Generating pneumatic pressure typically requires a large, noisy compressor; however, this challenge can be overcome in industrial settings by utilizing central pneumatic pressure systems, which are commonly found in factories and machine shops.

To effectively enhance endurance, the device must work in harmony with the user’s own muscles. Therefore, the control system utilizes an electromyographic (EMG) sensor to measure the proportional activation level of the muscles [15,16], along with pressure sensors to enable closed-loop control of the pneumatic muscle’s internal pressure. The experiments are conducted to determine the desired internal pressure based on the EMG sensor’s measurements of the muscle activation voltage. A PID control algorithm [17] is developed with feedback from pressure sensors. The overall control method can be viewed as a combination of model-based and PID control techniques.

This paper discusses the design and creation of a working prototype for a personal protection device, demonstrating the conceptual feasibility of using a wearable soft robotic arm assistance device to enhance lifting endurance and strength. It begins with several isolated component tests to guide the selection and design of key components and determine essential parameters for the control system. Additionally, the development of the prototype and the PID control algorithm is detailed. Procedures for the testing setup, results, and the evaluation of its functionality are presented in a later section.

## 2. Materials and Methods

A working prototype was developed to investigate the feasibility of whether artificial muscles could enhance human endurance during repetitive lifting tasks by attaching them to the user, alongside their natural biological muscles, and controlling them to nearly instantaneously mimic the actions of those corresponding biological muscles. To prove the concept, the prototype was designed to enhance only a single arm muscle. The biceps brachii was chosen as the target muscle, due to its relatively large size, easy accessibility, and its primary role in lifting movements. To replicate the actions of the user’s biceps, a closed-loop control system was developed. When the user lifts a weight, their bicep contracts. The EMG sensor measures the bicep’s activation voltage, which the controller uses to estimate the bicep’s proportional degree of activation, ranging from no exertion to maximum exertion. The controller then leverages a static model and pressure sensor feedback to create an equivalent proportional degree of activation in the pneumatic muscle by adjusting its internal pressure between atmospheric pressure and maximum supply pressure. The user can then modify the activation of their bicep based on the force they perceive the pneumatic muscle exerting on their arm. By imitating the activation of the corresponding biological muscle, the device can enhance the bicep, while being intuitively controlled by the user.

As a wearable enhancement device designed for industrial workers, several additional objectives were considered important to the design of the prototype. The device should work in relative unison with the user’s own muscles, which requires an activation delay that is small enough not to be noticeable to the user. This threshold is subjective, but 100 ms [18] is a common approximation in the fields of game development and software engineering. Ideally, the device should not limit the user’s range of motion or flexibility. Additionally, the device should be safe for users and reasonably priced.

A soft actuator as a pneumatic muscle, a control system, and a method for attaching the pneumatic muscle to the human arm were developed to achieve the primary objective, while addressing additional design goals. In the following sections, Section 2.1 discusses the various designs of pneumatic soft muscles and their testing, as well as how the final design of the pneumatic muscle was selected. Section 2.2 focuses on pressure control strategies and methods, while Section 2.3 discusses the use of EMG sensors to establish the relationship between EMG measurements and lifting weights. Section 2.4 describes how a model of the relationship between the valve duty cycle and the resulting pressure inside the pneumatic muscle was developed, based on testing the pneumatic actuator, and Section 2.5 examines how various PWM frequencies influence the pressure ratio across a range of duty cycles. Finally, Section 2.6 examines the effects of varying the number of valves and different muscle lengths on the system’s ability to achieve the target pressure and maintain quick response times.

### 2.1. Pneumatic Soft Muscle

Two crucial structural components of a pneumatic muscle are the inner latex tube and the surrounding mesh. The principle behind a pneumatic muscle is that, with a fixed volume, its length decreases as the cross-sectional area increases [12], as illustrated in Figure 1a. The total volume of the flexible tube acts as this fixed volume, while the mesh encasing the tube restricts the direction and magnitude of dimensional changes. For this project, rubber tubes and woven nylon expandable wire sheathing were chosen to build the pneumatic muscle. These materials were selected due to their low cost, high availability, and typical usage in pneumatic muscles.

Key parameters that significantly impact the performance of a pneumatic muscle are the inner diameter (ID) of the rubber tube, the tube’s thickness, the diameter of the mesh, and the muscle’s overall length.

The ideal actuator for bicep enhancement should be optimized to achieve a balance between maximizing the pulling force, enhancing proportional contraction, and minimizing the overall length. These characteristics are essential in a pneumatic muscle for this application because the enhancement needs to provide a full range of motion, while offering substantial assistive strength within the limited volume of a wearable device. To optimize the design of the prototype’s actuator according to these criteria, four pneumatic muscles with varying parameter values were constructed for testing, as shown in Figure 1b. Their parameter values are listed in Table 1.

To determine the optimal muscle and its characteristics, the performance of each muscle was tested, using the testing frame shown in Figure 2. A clamp secured the top of the muscle, while a fixed measuring tape assessed the muscle’s contraction. The wooden frame was positioned on a flat floor, and an air compressor actuated the pneumatic muscles under various loads.

Each pneumatic muscle was actuated with pressure ranging from 30 psi to 90 psi, in increments of 10 psi. The pressure was provided with a portable 3 gallon max 100 psi air pressure tank. At each pressure level, the muscle was tested with weights attached to the end, varying from 5 to 20 lbs, in increments of 5 lbs. The muscle’s contractions (or strain) were recorded for every combination of applied pressures and weights. The test results are plotted on a graph, as shown in Figure 3. The contractions of Muscles #3 and #4 were minimal compared to those of the first two test muscles. It was evident that pneumatic muscles with thinner walled tubes (Muscles #1 and #2) significantly outperformed those with thicker walled tubes (Muscles #3 and #4) across the relevant range of pressures in terms of contractions. Muscle #2 demonstrated a higher proportional contraction than muscle #1 across all combinations of hanging load and supply pressure. Muscle #3 exhibited a significantly lower proportional contraction under each condition compared to the first two muscles. The performance of muscle #4 closely mirrored that of muscle #3. For clarity and easy comparison, only a single dataset from the testing of Muscles #3 and #4 is included in Figure 3. In the figure, muscle #2 distinguished itself as the superior option among the four, owing to its equivalent strength and enhanced proportional contraction. Therefore, the optimum pneumatic muscle for the soft actuator in the working prototype should incorporate muscle #2’s parameters, including an inner diameter of 1/2 inch, a rubber tube thickness of 1/16 inch, and a mesh diameter of 1/2 inch.

### 2.2. Pressure Control System

Once the pneumatic soft muscles were determined as the actuators for the prototype, a method for controlling their contraction had to be established. The desirable characteristics of this control system include intuitive operation, rapid response, and robust control. Since the pressure within the soft muscle can be measured more conveniently and accurately than the assistance force it exerts, it was chosen as the operative variable for the control system. This choice is supported by bench tests, indicating that the pressure within the actuator serves as a reliable proxy for the force it provides. Key characteristics of the pressure-regulating hardware include a high flow rate, a quick response time, and control authority across the entire range of pressures that the actuator can effectively command. Several alternatives were examined for physically regulating the pressure within the soft muscle. The most expensive option was an integrated digital pressure regulator, which offers the fastest response times, the highest flow rates, and the best pressure control. Servo-actuated proportional valves represent the most cost-effective solution; they can be 3D printed from open-source designs and equipped with two-way valves, requiring additional valves to vent pressure from the actuator.

Ultimately, PWM (pulse width modulation)-driven three-way solenoid valves were chosen for their ability to occupy a more optimal position between the two extreme alternatives. Their response times fall within the millisecond range, and they can be connected in parallel to enhance the flow rate. When activated, they connect the actuator to the supply pressure, and when deactivated, they allow the actuator to vent pressure into the atmosphere. Although the activation of a solenoid valve is binary, pulse width modulation is used to cycle it rapidly. By varying the time spent activated versus deactivated in each cycle, the actuator can theoretically maintain any arbitrary pressure between the supply pressure and atmospheric pressure at the manifold’s exhaust ports. The specific model and configuration of the solenoid valves used were inspired by Soft Robotics Toolkit’s open-source fluidic control board design [19]. Their combination of a quick response, flexibility, and reasonable price range makes solenoid valves the optimal choice for inclusion in the pressure control hardware for the prototype.

Figure 4 depicts the muscle pressure control strategy. Muscle pressure is regulated using PWM across four parallel, three-way normally closed solenoid valves. An Arduino Mega 2560 microcontroller generates the appropriate PWM signals to operate the valves by processing input from the EMG and pressure sensors via the control algorithm.

The muscle pressure control algorithm determines how the prototype behaves in response to the user’s control commands. In this instance, the EMG measurement is regarded as the control command (or input), based on the assumption that it reflects the user’s bicep exertion level. The pneumatic muscle’s internal pressure is viewed as the system’s response (or output), under the assumption that it correlates with the pneumatic muscle’s actuation force. An empirical relationship translates the measured EMG voltage into the desired pressure. For this proof-of-concept testing, a linear relationship was established so that the pneumatic muscle’s proportional exertion equaled the approximate proportional exertion of the user’s bicep. This relationship could be modified to offer the user more or less assistance, or even varying levels of assistance, depending on the degree of bicep activation. The desired pressure command is passed to a model-based controller and a PID loop that utilizes pressure sensor feedback. The model-based controller outputs a target duty cycle intended to achieve the desired pressure. This target duty cycle is then adjusted by adding the duty cycle correction factor generated by the PID loop. This hybrid control scheme enables the PID loop to compensate for dynamic conditions not accounted for in the model, thereby enhancing the prototype’s responsiveness. The model-based controller adds value by serving as an experimentally verified baseline for stable, yet slow, controllability. Without the model-based controller offering a reasonable starting point, tuning a standalone PID controller would have required many more iterations. The corrected target duty cycle is then forwarded to the microcontroller’s PWM generation function, which produces the signal to actuate the valves, with the appropriate duty cycle to generate the desired internal pressure.

### 2.3. Muscle Contraction Sensor and Activation with Weight Lifted

An EMG muscle contraction sensor was chosen for the wearer to input commands into the control system. EMG is a technique for assessing the activation level of a biological muscle by measuring slight variations in the electrical potential that occur during muscle contraction. It was hypothesized that EMG measurement approximates the proportional activation force exerted by the bicep, as the force exerted is related to the lifted weight; consequently, an increase in the weight lifted would result in a higher EMG reading. Proving this was essential, as the EMG sensor was to serve as the control input. Additionally, obtaining a model for the relationship between the EMG readings and the weight lifted was also of interest, as it would be useful for determining how much assistance the system should provide.

An EMG sensor was attached to the subject’s arm, according to the sensor’s specifications and connected to an Arduino Mega, which was programmed to output the sensor’s readings to its serial port. In this research, a MyoWare Muscle Sensor [20] was selected as the EMG sensor, due to its simplified setup and calibration procedure. The sensor includes onboard signal filtering and integration, as well as an onboard gain adjustment potentiometer. It outputs a voltage that is a proportion of the microcontroller’s operating voltage. The Arduino Mega features a 10-bit ADC, providing values ranging from 0 to 1024. The Arduino Mega’s serial port was connected to a laptop via a USB, and MATLAB 9.6 (2019a) was used to collect and display the data from the serial port in real time. To prevent clipping, the sensor’s gain was adjusted using its onboard potentiometer, ensuring that the subject’s maximum bicep contraction during flexing resulted in an output below the sensor’s maximum threshold.

The subject, a 23-year-old male, performed three consecutive seated dumbbell curls, followed by a partial curl, which was held at the top for about four seconds. This was repeated with six different weights, increasing by five pounds, from zero to twenty-five pounds. The subject rested for five minutes between trials to avoid the impact of fatigue on the lifting results. During each trial, a mark on the user’s bicep was used to ensure the EMG was placed in the same spot, maintaining consistency in the reading results. The results of the six trials are illustrated in Figure 5a, which shows a clear increase in the EMG readings as the weight lifted increases. Figure 5b displays a graph illustrating the correlation between the amount of weight lifted and the maximum activation measurements obtained from each trial’s EMG reading dataset. The findings suggest a predominantly linear relationship between these two variables.

A method for directly measuring the bicep’s activation force was not explored or available, so these results are used to indirectly verify that the EMG reading is approximately proportional to the intensity of the bicep’s contractions at any given moment. According to Newton’s second law, it is clear that the force exerted by the bicep to overcome the gravitational pull on the weight and generate a specific upward acceleration must increase in proportion to any rise in the weight’s mass. Since the test subject attempted to lift each weight with a consistent path and speed, variations in muscle force caused by changes in lifting technique should be minimal. Therefore, in the absence of overexertion, the peak muscle force during each trial should rely on the mass of the weight being lifted. If the peak muscle force exerted is proportional to the weight lifted and the peak EMG reading is also proportional to the weight lifted, then the peak EMG reading will correlate with the peak muscle force exerted. Figure 5b indeed shows that the weight lifted and the peak EMG readings are proportional, suggesting that the EMG reading and bicep contraction force can be assumed to be proportional.

The findings presented, particularly illustrated in the graph representing the 20 pound test, reveal that the sensor effectively identifies a distinct increase in effort correlated with muscle fatigue. To validate this observation, an additional experiment was conducted involving twenty consecutive curls at a weight of twenty pounds, as depicted in Figure 6.

Indeed, the first few repetitions show an increase in effort, while additional repetitions begin to experience clipping, and the time spent at the maximum reading continues to rise, indicating that progressively larger portions of each spike are being clipped. This trend appears to persist with increasing repetitions. The results shown in Figure 6 suggest that as the bicep tires, more effort is required to maintain consistent performance. The shape and minimum EMG reading of each cycle appear to be independent of muscle fatigue. Although the control algorithm does not compensate for this clipping, the controller still activates the pneumatic muscle maximally for the clipped peaks, which is the desired response, even if those peaks are recorded. Since this peak clipping occurs after prolonged activity and does not disrupt the proper functioning of the prototype, it was deemed acceptable to regard this as insignificant for the purpose of creating a working prototype to demonstrate the basic feasibility of EMG-controlled soft robotic muscle enhancement.

### 2.4. Output/Supply Pressure Ratio and Valve PWM Duty Cycle

Determining the relationship between the output/supply pressure ratio and the valve PWM duty cycle was essential for controlling the output pressure. Static tests were conducted to establish this relationship. Test muscle #2 (the selected muscle for the prototype) was clamped to the edge of a table at one end, with a five pound weight attached to the other end. The control system was setup as illustrated in Figure 4. The Arduino Mega was programmed to increase the valve PWM duty cycle in increments of 1%, pause for two seconds to allow the system to stabilize, and subsequently transmit data from the pressure sensors to the Arduino IDE serial monitor, along with the current duty cycle. Two hundred data points were collected in quick succession at the conclusion of each duty cycle increment, and the pressure data for each duty cycle percentage increment was averaged and documented. The ratio of the supply pressure to the output pressure against the duty cycle was then plotted, as illustrated in Figure 7a. While the small upper and lower regions of the graph remain constant, the central section exhibits a prominent quadratic relationship. This observation is verified by using MATLAB’s polyfit() function [21], which fits a second-order polynomial to the dataset. The resulting curve, overlaid on a plot of the average data, is shown in Figure 7b. Therefore, the appropriate duty cycle range, PWM frequency, and quadratic fit coefficients of the system can be determined based on the outcomes of these experiments or from Figure 7b. With the model obtained using the polyfit() function and applying the quadratic formula, the duty cycle required to achieve a certain target pressure can be calculated by equating the target pressure ratio (target pressure divided by the current supply pressure reading) to the model obtained from polyfit() or by using the quadratic formula.

Following the selection of muscle #2 (12 inches in length) for the prototype, the actual length of the muscle was determined by considering anthropometric data (adult men’s body size) for ergonomic purposes. A customized muscle length of 27 inches was established for the prototype. The previous test was repeated using a 27-inch-long muscle. This produced another interesting result: the larger volume of the new muscle smoothed out some irregularities observed in the data from the 12-inch-long version. This result is illustrated in Figure 8, which shows that a longer muscle is preferred.

### 2.5. Determine the PWM Frequency

Initially, it was assumed that a higher PWM frequency would enable more accurate and precise control of the pressure ratio. To determine if this was true, the previous test, which measured the pressure ratio across a range of duty cycles, was repeated multiple times at different PWM frequencies. The experimental results are shown in Figure 9a, which illustrates an increased range of usable duty cycles as the PWM frequency decreases. For instance, at a PWM frequency of 100 Hz, the duty cycle range is from 13% to 68%, while at a PWM frequency of 25 Hz, the duty cycle range extends from 3% to 94%. This suggests that a lower frequency would be preferable, as it permits somewhat finer control. To further verify this, each dataset is fitted with the quadratic formula, and the mean squared error for each fitted curve model is calculated at various PWM frequencies. A plot of the model error at each frequency is presented in Figure 9b, indicating that there is a limit to how much the frequency can be reduced before the model’s reliability begins to be negatively affected.

The increased error (as shown in Figure 9b) and reduced effective duty cycle range (as shown in Figure 9a) at higher frequencies are likely due to the response time of the valves. According to the catalog for the valves, they take 3.5 ms to switch to their on (open) state and 2 ms to switch back to their off (closed) state. At the higher tested frequencies, the time required for the valves to fully switch states occupies a significant portion of the PWM period (for instance, the period is only 10 ms at 100 Hz), resulting in the valves opening and closing more frequently. This more frequent switching disrupts the air flow more often, likely causing the valve to be partially open and partially closed simultaneously, which contributes to the increased error shown in Figure 9b. At either end of the duty cycle range for the higher frequency tests, the switching time exceeds the duration for which the valve is open or closed, leading to the valve effectively never being fully open or fully closed. This results in the differences in the effective duty cycle ranges shown in Figure 9a. At lower frequencies, the increasing error likely stems from the greater magnitude of pressure oscillations facilitated by the longer duration, which aligns with the initial hypothesis. Considering Figure 9a,b, for this application, a frequency of approximately 45 Hz appears to be optimal for minimizing model error, while also providing one of the widest usable duty cycle ranges among the tested frequencies.

### 2.6. Response Time, Number of Valves, and Optimal Muscle Length

The effects of varying numbers of valves and different muscle lengths on the system’s ability to reach a target pressure over a short time interval were tested by modifying the controller codes to produce a low-frequency square wave output, which helped determine the system’s approximate step response. This operated the valves at a 100% duty cycle for a short duration (e.g., 1 s, 0.5 s, 0.25 s, etc.), then switched them to a 0% duty cycle for the same duration before returning to 100%, and so forth. Data from the pressure sensors were output to the Arduino IDE’s serial monitor for observation. Although the response time was not directly measured, comparing the magnitudes of the pressure changes over a set time interval allowed for inferences about the general effects of adding valves and increasing the muscle length. Running additional valves in parallel caused the muscle’s pressure to drop closer to zero at a 0% duty cycle, allowing it to approach the input pressure at a 100% duty cycle. In contrast, increasing the muscle’s length and, consequently, its volume produced the opposite effect. Therefore, it can be concluded that adding valves enables the system to respond more quickly, while increasing the muscle length slows the system’s response. These results guided the final system design, suggesting that the pneumatic muscle should be only as long as necessary to achieve the desired contraction length, and that as many valves as reasonably possible should be utilized to maximize the potential air flow rate into and out of the muscle.

Regarding the final muscle length, it was determined by evaluating its impact on the response time (shorter is better) and smoothing out the irregularities discussed in Section 2.4 (a longer muscle is preferred, as it smooths out some irregularities in the data curve of the output/supply pressure ratio and valve PWM duty cycle, as illustrated in Figure 8). The optimal muscle length was then further considered in relation to ergonomic factors to accommodate the general population of workers in the manufacturing sector, by reviewing and utilizing their anthropometric data. A customized muscle length of 27 inches was ultimately established for the prototype. It was later justified during system testing that this length ensures user comfort, ensures the muscle remains compact, and provides a reasonable response time for efficiency purposes.

Regarding the number of valves in the prototype, four solenoid valves was identified as the optimum, balancing the need to maintain the device’s compactness, while also reducing the soft actuator’s response time by increasing the flow rate to and from the actuator.

## 3. Results and Discussion

This section presents the final configuration of the prototype system, the experimental setup, which involved two subjects wearing the device, the test results, and their interpretation. These test results were used to confirm that the EMG-controlled soft robotic bicep enhancement is an effective method for increasing the effective endurance of the bicep muscle.

### 3.1. Final Configuration of the Control System

The final configuration of the control system for the prototype consists of three interconnected subsystems. The pneumatic subsystem comprises a pressure supply, a soft actuator, pressure sensors, and a manifold of solenoid valves that regulate the pressure within the muscle actuator. The electronic subsystem includes the EMG and pressure sensors, the valves, along with their driving MOSFETs, an Arduino microcontroller, and appropriate power supplies for these components. The software system contains the control algorithms running on the microcontroller. These algorithms read the sensors and instruct the microcontroller to send PWM signals to the valve-driving MOSFETs. These electronic, pneumatic, and software components work in concert to ensure the soft actuator functions in harmony with the wearer’s bicep.

The architecture of the electronic subsystem is illustrated in Figure 10. All its components are ultimately powered by a 30 amp capacity 12 volt direct current power supply that accepts a standard US residential alternating current. However, no component directly uses the 12 volt power, so two adjustable direct current regulators are used to provide 9 and 24 volt power, respectively. The 9 volt regulator powers the Arduino Mega microcontroller, which provides five volt power to all the sensors, measures their output, and sends PWM signals to the MOSFETs. The four solenoid valves were powered by a 24 volt regulator and switched on and off by a set of four power MOSFETs, which can be thought of as solid state relays. These MOSFETs, or metal oxide–semiconductor field-effect transistors, are located on a board that optically isolates the two voltages allowing five volt PWM outputs from the Arduino to rapidly switch the 24 volt power. This allows 24 volt PWM input signals to be sent to each valve from the Arduino, despite the microcontroller’s five volt maximum output.

The pneumatic subsystem of the controller supplies the energy that actuates the soft muscle via its connection to a source of pressurized air. It enables the Arduino Mega to control the pressure within the soft actuator by operating the four solenoid valves. This is illustrated in Figure 11. For safety and convenience, a ball valve can be used to manually isolate the pneumatic system from the pressure supply. The supply pressure is routed to the four valves through the manifold, which also connects the valves to the atmosphere via exhaust ports. The four valve outputs of the manifold converge within the end cap, before entering the interior of the soft actuator. Pressure sensors are integrated into the system at the points where the supply pressure enters the manifold and where the valve output exits the manifold. A third pressure sensor is exposed to the atmosphere. These pressure sensors measure the supply pressure, the pressure within the soft actuator, and atmospheric pressure. Although the readings from the atmospheric pressure sensor are not utilized by the control algorithm, they enable the manual verification of the functionality and accuracy of the other two identical gauge pressure sensors during calibration. Four parallel solenoid valves were used to reduce the soft actuator’s response time by increasing the flow rate to and from the actuator.

The software program running on the Arduino Mega reads the EMG and pressure sensors, implements a control algorithm to determine the appropriate output, and generates the necessary output. The Arduino’s built-in functions and various open-source libraries managed the intricate details of reading the sensors and performing PID calculations, while the structure and gains of the control algorithm were specifically developed and tuned for the custom soft actuator used in the prototype. As illustrated in Figure 4, the control algorithm is a hybrid of the empirical control relationship determined through static testing and a typical PID control loop. A cycle of this control loop begins with a reading from the EMG sensor. After passing through a moving average filter, the EMG signal is used to calculate the desired actuator pressure using a linear relationship qualitatively approximated from the user feedback and sensor testing (Figure 5b—Linear relationship between the weight and EMG readings). This target pressure is considered in regard to the quadratic relationship between the pressure and duty cycle (Figure 7b—Pressure ratio and duty cycle), which is used to determine the duty cycle command to produce the target pressure, considering the measured supply pressure. This relationship was determined under static conditions with a constant load and does not account for dynamic conditions and load variations that occur during the actual use of the prototype. During testing, it was determined that these factors needed compensation to achieve an acceptable level of performance. Therefore, a PID controller, based on the actuator’s pressure, was used to introduce a correction factor to the duty cycle command. The PID controller takes the error between the target pressure and the most recent measurement of the pressure inside the actuator and acts on it to produce a corrective change in the duty cycle command. The size of the correction is constrained to remain within a specific proportion of the initial duty cycle command, ensuring that the empirical model continues to serve as the dominant controller. After the PID controller gains underwent several rounds of manual tuning, qualitative comparisons by a test wearer indicated that the addition of closed-loop control made the actuator feel more responsive and assistive. In the final stage involving the control algorithm, the corrected duty cycle command generates a corresponding PWM output to the valves at a specified PWM frequency.

### 3.2. Physical Setup

With the parameters of the pneumatic muscle and control system configured, the wearable device prototype was built. The wearable device consists of a pneumatic muscle, an EMG sensor, a control box, an elbow anchor, a harness with an adjustable buckle, and a power supply, as illustrated in Figure 12 and Figure 13.

The shoulder harness was designed based on a shoulder brace and pauldron design, using nylon straps and rivets. An elbow brace, also made of nylon, attaches to the forearm and connects the soft pneumatic muscle through the shoulder to the controller, located at the back. The control box was 3D printed to accommodate the electronics and control components. The interior of the control box is illustrated in Figure 14. The air supplied to the control box comes from the pneumatic system in the building (UC Makerspace: a 12,000 square foot fabrication and maker space), with a pressure range of 90–100 psi.

### 3.3. Results

Two 23-year-old males (User A and User B) were recruited to lift weights, both with and without assistance from the wearable prototype. User A’s two trials took place on consecutive days. In Trial #1, User A lifted a 20 pound weight without using the assistance device, and he managed to do so 20 times before stopping. After one hour of rest, User A repeated the lifting exercise while wearing and using the assistance device. This time, User A was able to lift the 20 pound weight 35 times. So, in Trial #1, User A’s lifting endurance increased by 75% with the aid of the device. The next day, User A completed Trail #2 by repeating the practice from Trial #1. This time, he was able to lift the 20 lbs weight 22 times without the assistance device and 37 times with it. His endurance increased by 68% in Trial #2. On average, in the first and second trials by User A, the device increased User A’s maximum lifting endurance by 71.59%.

User B’s trials were conducted on the same day due to time constraints. In the third trial, User B, who is not as strong as User A, lifted a weight of 20 lbs 14 times, both with and without the assistance device. For Trial #4, a more comfortable weight was selected for User B, and it began immediately after the completion of Trial #3, with no pause between the two trials. This time, User B lifted the 15 pound weight without assistance eight times before failing. Then, the prototype was activated, and User B lifted the weight 18 times. User B’s maximum lifting endurance increased by 125% when using the assistance device, even without significant recovery time. The test results are summarized in Table 2.

The feasibility test data presented in Table 2 indicate that a wearable soft robotic bicep augmentation could enhance a user’s endurance for lifting tasks within their lifting capability (Trials #1, #2, and #4). However, there is no evidence of an endurance increase (Trial #3) when the lifting tasks exceed their lifting capability. Neither test user reported any significant delays in response or limitations in regard to their range of motion.

## 4. Conclusions

A soft robotic arm assistance device that mimics the user’s muscle anatomy and activation states could serve as a viable solution for enhancing the endurance of individuals who frequently lift weights, particularly assembly workers in the manufacturing sector. The device integrates a McKibben-inspired pneumatic muscle with an EMG-controlled closed-loop pressure control algorithm. The prototype of the bicep assistance device discussed in this paper is sufficiently robust to assist various users and responsive enough to prevent noticeable lag. The testing involving a sample user suggests that the prototype’s concept is feasible. The major innovative feature can be recognized as the device’s flexibility and wearability, which allow it to fit different users, as well as its effectiveness in enhancing the user’s bicep strength and endurance.

## Figures and Tables

**Figure 1 bioengineering-12-00526-f001:**
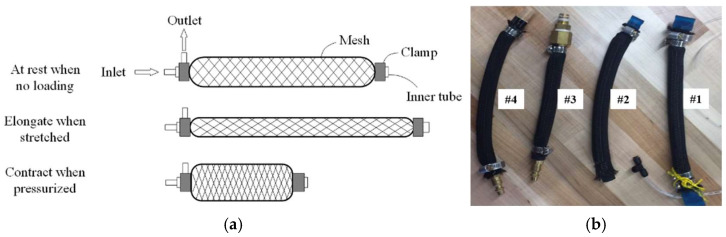
**Pneumatic muscle**: (**a**) working principle; and (**b**) four prototypes labeled #1 to #4.

**Figure 2 bioengineering-12-00526-f002:**
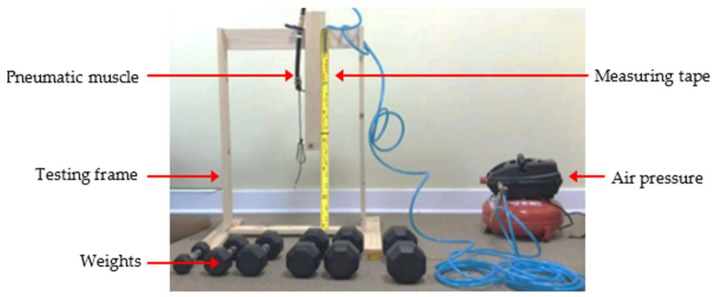
Pneumatic muscle testing platform and setup.

**Figure 3 bioengineering-12-00526-f003:**
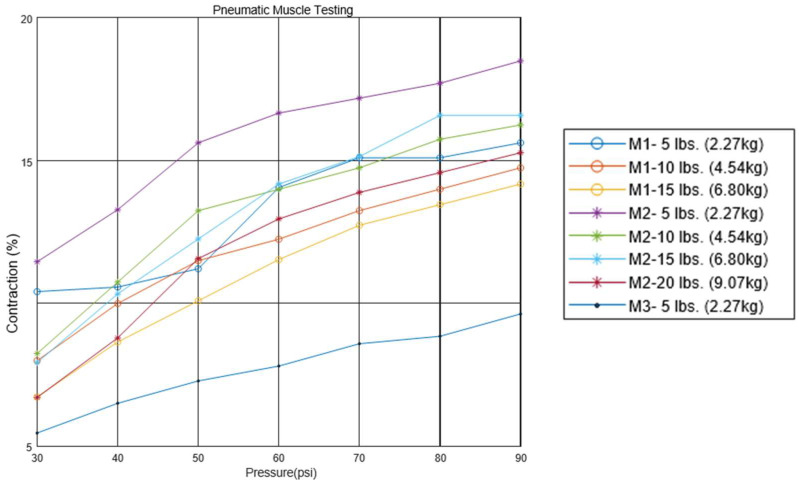
Relationship between pressure, muscle contraction, and loads from muscle tests (Note: M1–5 lbs (2.27 kg) refers to muscle #1 with a 5 lbs (2.27 kg) load).

**Figure 4 bioengineering-12-00526-f004:**
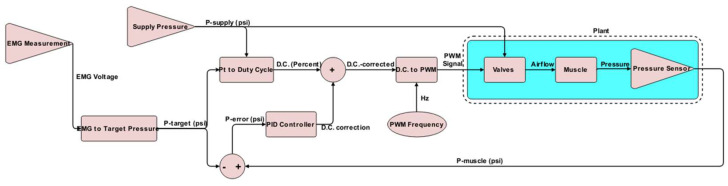
Muscle pressure control algorithm flowchart.

**Figure 5 bioengineering-12-00526-f005:**
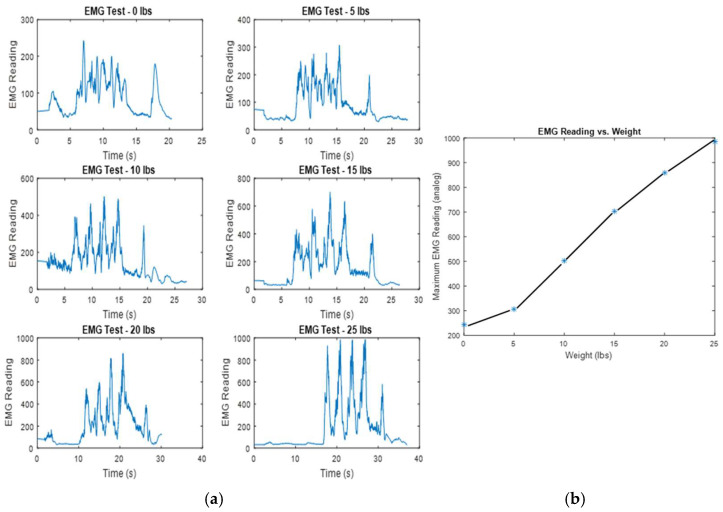
(**a**) Test datasets of EMG readings and various lift weights; and (**b**) the relationship between the weights lifted and maximum EMG readings (* indicates the maximum EMG reading from the test dataset at a specified weight).

**Figure 6 bioengineering-12-00526-f006:**
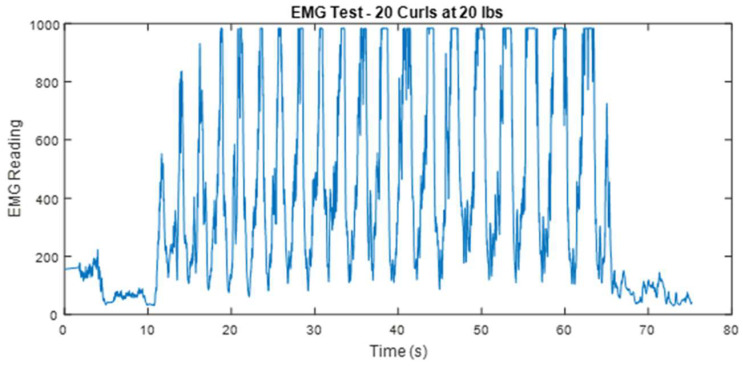
EMG readings for 20 repetitions with a lift weight of 20 lbs.

**Figure 7 bioengineering-12-00526-f007:**
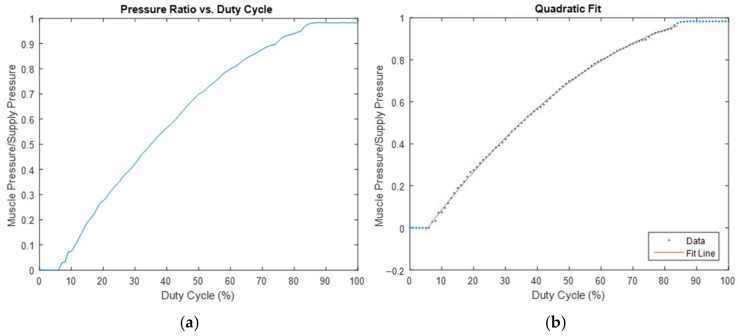
(**a**) Test results on relationship between pressure ratio and duty cycle; and (**b**) quadratic fit of pressure ratio–duty cycle relationship overlaid on the test data.

**Figure 8 bioengineering-12-00526-f008:**
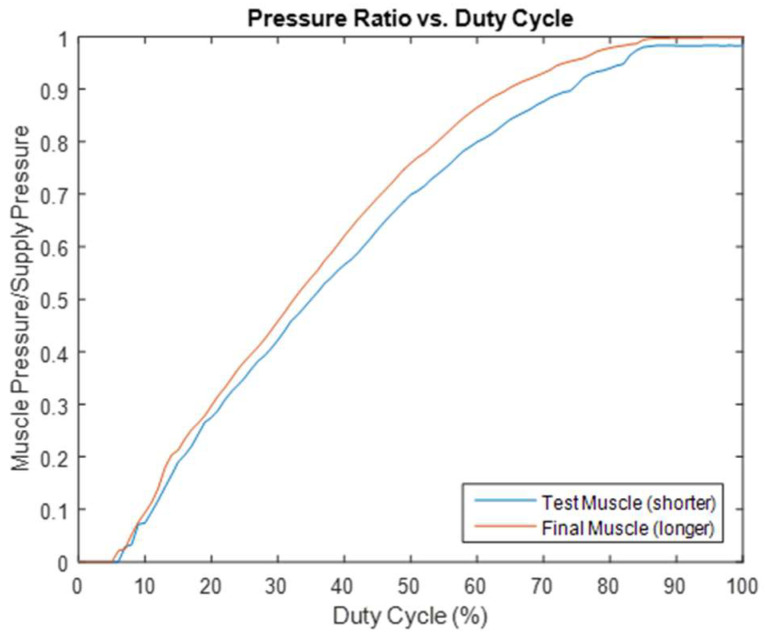
Effect of muscle length on the smoothness of the pressure ratio–duty cycle curve.

**Figure 9 bioengineering-12-00526-f009:**
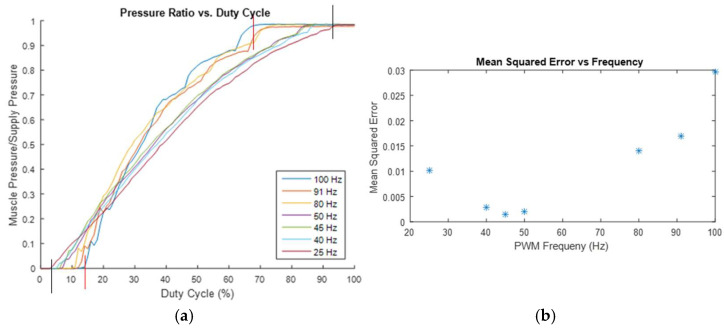
(**a**) Pressure ratio vs. duty cycle curves for various PWM frequencies (two red vertical lines indicate the duty cycle range at 100 Hz PWM frequency and two black vertical lines indicate the duty cycle range at 25 Hz PWM frequency); and (**b**) mean squared error of quadratic fit at various PWM frequencies (* indicates mean squared error at specified PWM frequency (Hz)).

**Figure 10 bioengineering-12-00526-f010:**
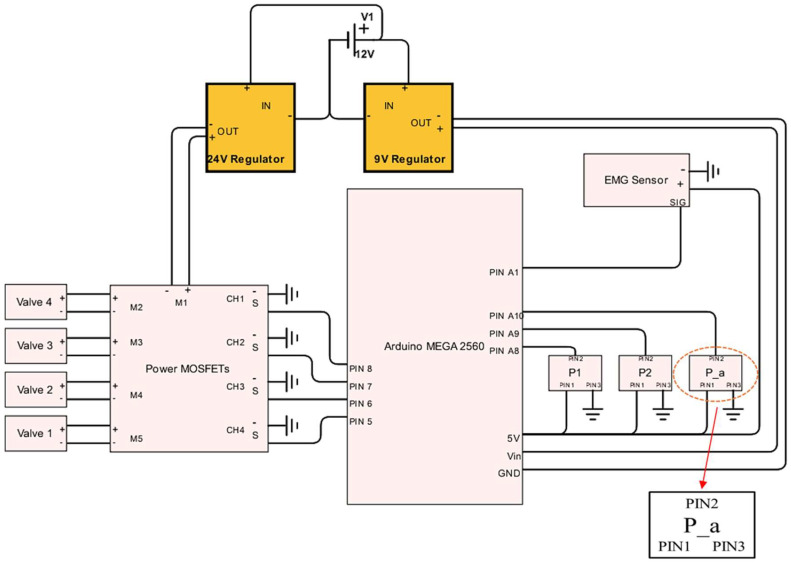
Control system electronic wiring diagram.

**Figure 11 bioengineering-12-00526-f011:**
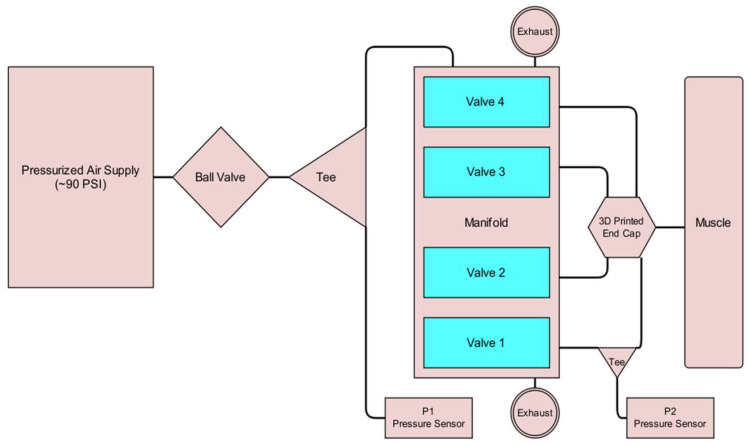
Pneumatic control system diagram.

**Figure 12 bioengineering-12-00526-f012:**
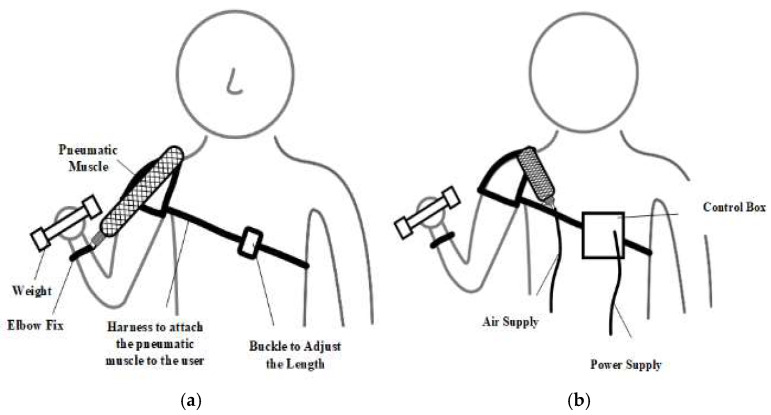
Wearable device prototype testing setup illustration: (**a**) front; and (**b**) back.

**Figure 13 bioengineering-12-00526-f013:**
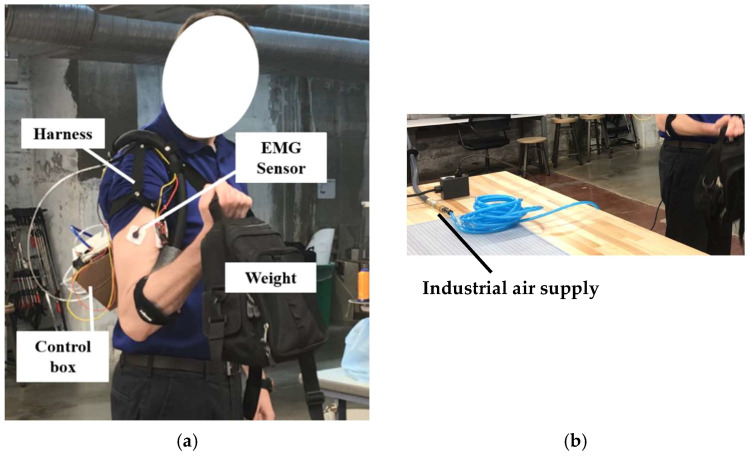
(**a**) User wearing physical prototype for testing; and (**b**) industrial air supply to the prototype.

**Figure 14 bioengineering-12-00526-f014:**
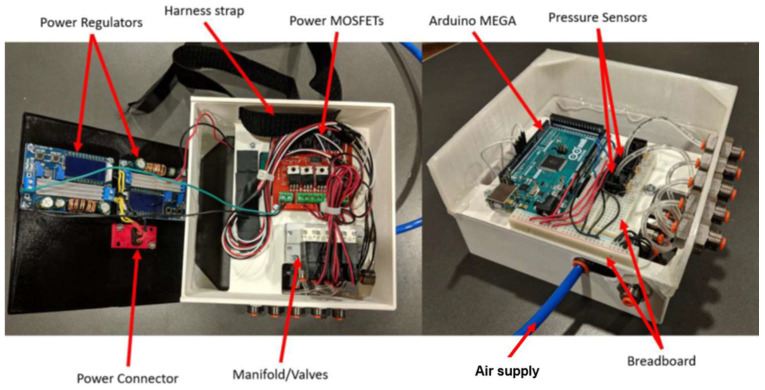
Control box with electronic components and controls.

**Table 1 bioengineering-12-00526-t001:** Four muscles’ parameters and values.

	Rubber Tube (in)			Mesh (in)
Test Muscle #	ID	Thickness	Length	Diameter
#1	1/2	1/16	12	1/4
#2	1/2	1/16	12	1/2
#3	1/2	1/8	12	1/4
#4	1/2	1/8	12	1/2

**Table 2 bioengineering-12-00526-t002:** Test results on lifting capacity and repetitions.

Trial	User	Lifting Weight	Repetitions Without Wearing Prototype	Repetitions While Wearing Prototype	Endurance Increase
#1	A	20 lbs	20	35	75%
#2	A	20 lbs	22	37	68%
#3	B	20 lbs	14	14	0%
#4	B	15 lbs	8	18	125%

## Data Availability

The original contributions presented in this study are included in the article. Further inquiries can be directed to the corresponding author.

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
