# Peer review of "EMG-Controlled Soft Robotic Bicep Enhancementâ€"

_bioengineering, 2025, doi:10.3390/bioengineering12050526_

Round 1
Reviewer 1 Report
Comments and Suggestions for Authors
The paper topic is rather interesting - it is about artificial muscles and method of their testing. The muscle design and functioning is rather correct. However method of its testing seems to be very subjective. A person without described his power characteristics in one experiment lifts the load a few times, then he seems to be amplified by using the artficial muscle, he lifts the load more times. All the conclusions seem to be very subjective withot quantitative measurements of persons power without and with device. Nobody knows are the persons fatigue, how their relaxation impacts their power? In literature shold be descroptions of such evaluation for sportsmen. Maybe sportive methods of power evaluation could be useful for the topic applied.
Author Response
Response: Thank you for your comments. We will do our best to address your concerns.
The testing method described in the paper is our way of verifying our mechanism and hypothesis. We consider it straightforward and easier to conduct without using complex equipment. Section 3.3: Results briefly describe the person’s power. The conclusions are based on our test results. Instead of using sportive methods, we use an EMG sensor to evaluate the user’s lift capacity (power).
Reviewer 2 Report
Comments and Suggestions for Authors
The manuscript presents an elegant solution involving a robotic actuator and a control system designed to enhance endurance during lifting tasks by complementing bicep contractions. The authors tackle a critical issue – repeated heavy lifting by industrial workers – which can lead to serious arm injuries and accounts for over one hundred thousand musculoskeletal disorder cases annually in the U.S. alone. Thus, this paper is both timely and relevant, offering a system that could reduce injury risk by increasing muscular endurance and strength, potentially enhancing workplace productivity.
In contrast to existing rigid exoskeleton-based approaches, the study innovatively employs soft actuators (“artificial muscles”) that interface more naturally with the human body. A key contribution is the development of a closed-loop control system that correlates the internal pressure of pneumatic muscles with EMG sensor data, which captures the user's muscle activation. A working personal-use prototype has been built, demonstrating the feasibility of the approach to assist lifting tasks.
The manuscript is thorough and well-structured. It provides detailed descriptions of the experiments, starting with subsystem optimization – from pneumatic muscle design and pressure control to EMG-based sensing – and proceeding to full system integration and testing. The description is clear and leaves a positive impression. However, I have several concerns and suggestions for improvement:
- I recommended the change of terms: “muscle augmentation” commonly refers to the increase of apparent muscle volume, for instance, through surgical procedure in which fat is removed from one part of the body and placed into the muscles to augment their size. Therefore, it might be better to user other terms like, for example, Bicep Complementation (or Assistance, or Cooperation, or Enhancement) instead of Bicep Augmentation.
- Reference to Table 1 (line 139) appears to be broken. This should be corrected for clarity.
- Figure 3: I recommend updating the legend to explicitly include load units (e.g., “M1 – 5 lbs” instead of “M1-5”). Additionally, weights should preferably be listed in kilograms (or at least provide an equivalent), in line with standard scientific conventions.
- Figure 4: The image quality is noticeably low and should be improved for better readability and presentation.
- Line 242: “…MyoWare Muscle Sensor [20] was selected as the EMG sensor. It has a ten-bit resolution and outputs its readings as a proportion of the microcontroller’s operating voltage…” – I assume there is a confusion because 10-bit resolution refers to the Arduino’s ADC, not to the sensor itself.
- While the authors note that large, noisy compressors are typically required to generate pneumatic pressure and that centralized systems may be available in industrial settings, the pressure source used in the experimental setup is not described in sufficient detail. Additional information should be included: How large and noisy is the compressor used? Is it portable? Could it realistically be integrated with the wearable system in practice?
In summary, the manuscript presents a compelling and well-executed study on soft robotic muscle assistance, with significant potential for real-world applications. Addressing the concerns listed above would further strengthen the clarity and impact of the work.
Reviewer 3 Report
Comments and Suggestions for Authors
This work presents a soft robotic bicep augmentation system controlled by surface EMG signals. The design combines a McKibben pneumatic muscle, a closed-loop pressure control system, and a simple hybrid control algorithm. As a prototype study, the work is well executed, with clear methodology and reasonably complete testing. Experimental validation is sufficient for a proof-of-concept stage. There are two minor concerns about this study that authors need to further illustrate before considering publishing.
- EMG signals can vary even within the same person depending on fatigue or electrode placement.
- EMG amplitude differs significantly between individuals, yet the current system uses a fixed linear mapping with only two subjects.
The paper would benefit from a brief discussion on how calibration between EMG and pressure could be handled, either through additional experiments or theoretical suggestions.
